## OPEN

# SARS-CoV-2-specific T-cell epitope repertoire in convalescent and mRNA-vaccinated individuals

Julia Lang-Meli [1,2,12], Hendrik Luxenburger [1,2,12], Katharina Wild[1,3,12], Vivien Karl [1,4,12], Valerie Oberhardt[1,4,12], Elahe Salimi Alizei[1,3], Anne Graeser[1], Matthias Reinscheid[1,4], Natascha Roehlen[1], David B. Reeg[1], Sebastian Giese[5], Kevin Ciminski [5], Veronika Götz[1], Dietrich August[1], Siegbert Rieg[1], Cornelius F. Waller [6], Tobias Wengenmayer[7,8], Dawid Staudacher[7,8], Daniela Huzly[5], Bertram Bengsch[1,9], Georg Kochs [5], Martin Schwemmle [5], Florian Emmerich[10], Tobias Boettler [1,11,13], Robert Thimme[1,13 ✉], Maike Hofmann[1,13 ✉] and Christoph Neumann-Haefelin [1,13 ✉]

Continuously emerging variants of concern (VOCs) sustain the SARS-CoV-2 pandemic. The SARS-CoV-2 Omicron/B.1.1.529 VOC harbours multiple mutations in the spike protein associated with high infectivity and efficient evasion from humoral immunity induced by previous infection or vaccination. By performing in-depth comparisons of the SARS-CoV-2-specific T-cell epitope repertoire after infection and messenger RNA vaccination, we demonstrate that spike-derived epitopes were not dominantly targeted in convalescent individuals compared to non-spike epitopes. In vaccinees, however, we detected a broader spike-specific T-cell response compared to convalescent individuals. Booster vaccination increased the breadth of the spike-specific T-cell response in convalescent individuals but not in vaccinees with complete initial vaccination. In convalescent individuals and vaccinees, the targeted T-cell epitopes were broadly conserved between wild-type SARS-CoV-2 variant B and Omicron/B.1.1.529. Hence, our data emphasize the relevance of vaccine-induced spike-specific CD8+ T-cell responses in combating VOCs including Omicron/B.1.1.529 and support the benefit of boosting convalescent individuals with mRNA vaccines.

The SARS-CoV-2-specific T-cell epitope repertoire has been studied in some detail[1–6] (reviewed in Grifoni et al.[7]); however, comparative in-depth studies of the epitope repertoire targeted by infection- versus vaccine-induced T-cell responses are lacking thus far. Thus, precise prediction of the immune escape potential of emerging VOCs including Omicron/B.1.1.529 from the T-cell response in convalescent individuals compared to vaccinees is hardly possible. In this study, we evaluated SARS-CoV-2-specific T-cell responses in convalescent individuals recovered from SARS-CoV-2 infection ($n = 19$) as well as individuals after 2 ($n = 16$) and 3 ($n = 7$) doses of SARS-CoV-2 vaccination (Pfizer/BioNTech messenger RNA vaccine) (Supplementary Table 1).

### Results

**SARS-CoV-2-specific T-cell epitope repertoire.** We first mapped the overall SARS-CoV-2-specific CD8+ T-cell response against a set of 43 previously described immunodominant SARS-CoV-2-specific CD8+ T-cell epitopes (Supplementary Table 2) restricted by common human leukocyte antigen (HLA) class I alleles[1–6] in epitope-specific

T-cell cultures followed by interferon-γ (IFN-γ) staining. In agreement with their association with a mild COVID-19 course, CD8+ T-cell responses in convalescent individuals targeted most epitopes distributed over all viral proteins, with spike-specific epitopes not being dominant (Fig. 1a, left column). In vaccinees, in contrast and as expected, CD8+ T-cell responses were predominantly directed against spike epitopes (Fig. 1a, right column and Extended Data Fig. 1a). Few CD8+ T-cell responses targeted non-spike epitopes, with the HLA-B*07/N$_{105-113}$ epitope being the main target. For this epitope, cross-recognition by T cells against common cold coronaviruses has been suggested previously[8–10]. Individual spike-specific CD8+ T-cell epitopes were more often targeted in vaccinees compared to convalescent individuals; the spike-specific CD8+ T-cell repertoire also appeared broader in vaccinees compared to convalescent individuals. When we compared the corresponding viral sequences between wild-type (WT) SARS-CoV-2 variant B and Omicron/B.1.1.529, only a single tested optimal CD8+ T-cell epitope was affected by viral variation in subvariants BA.1 and BA.2 (Fig. 1a, red and Supplementary Table 3).

[1]Department of Medicine II (Gastroenterology, Hepatology, Endocrinology and Infectious Diseases), Freiburg University Medical Center, Faculty of Medicine, University of Freiburg, Freiburg, Germany. [2]IMM-PACT, Faculty of Medicine, University of Freiburg, Freiburg, Germany. [3]Faculty of Chemistry and Pharmacy, University of Freiburg, Freiburg, Germany. [4]Faculty of Biology, University of Freiburg, Freiburg, Germany. [5]Institute of Virology, Freiburg University Medical Center, Faculty of Medicine, University of Freiburg, Freiburg, Germany. [6]Department of Haematology, Oncology & Stem Cell Transplantation, Freiburg University Medical Center, Faculty of Medicine, University of Freiburg, Freiburg, Germany. [7]Department of Medicine III (Interdisciplinary Medical Intensive Care), Freiburg University Medical Center, Faculty of Medicine, University of Freiburg, Freiburg, Germany. [8]Department of Cardiology and Angiology I, Heart Center, Freiburg University Medical Center, Faculty of Medicine, University of Freiburg, Freiburg, Germany. [9]Signalling Research Centres BIOSS and CIBSS, University of Freiburg, Freiburg, Germany. [10]Institute for Transfusion Medicine and Gene Therapy, Freiburg University Medical Center, Faculty of Medicine, University of Freiburg, Freiburg, Germany. [11]Berta-Ottenstein Programme, Faculty of Medicine, University of Freiburg, Freiburg, Germany. [12]These authors contributed equally: Julia Lang-Meli, Hendrik Luxenburger, Katharina Wild, Vivien Karl, Valerie Oberhardt. [13]These authors jointly supervised this work: Tobias Boettler, Robert Thimme, Maike Hofmann, Christoph Neumann-Haefelin. ✉e-mail: robert.thimme@uniklinik-freiburg.de; maike.hofmann@uniklinik-freiburg.de; christoph.neumann-haefelin@uniklinik-freiburg.de

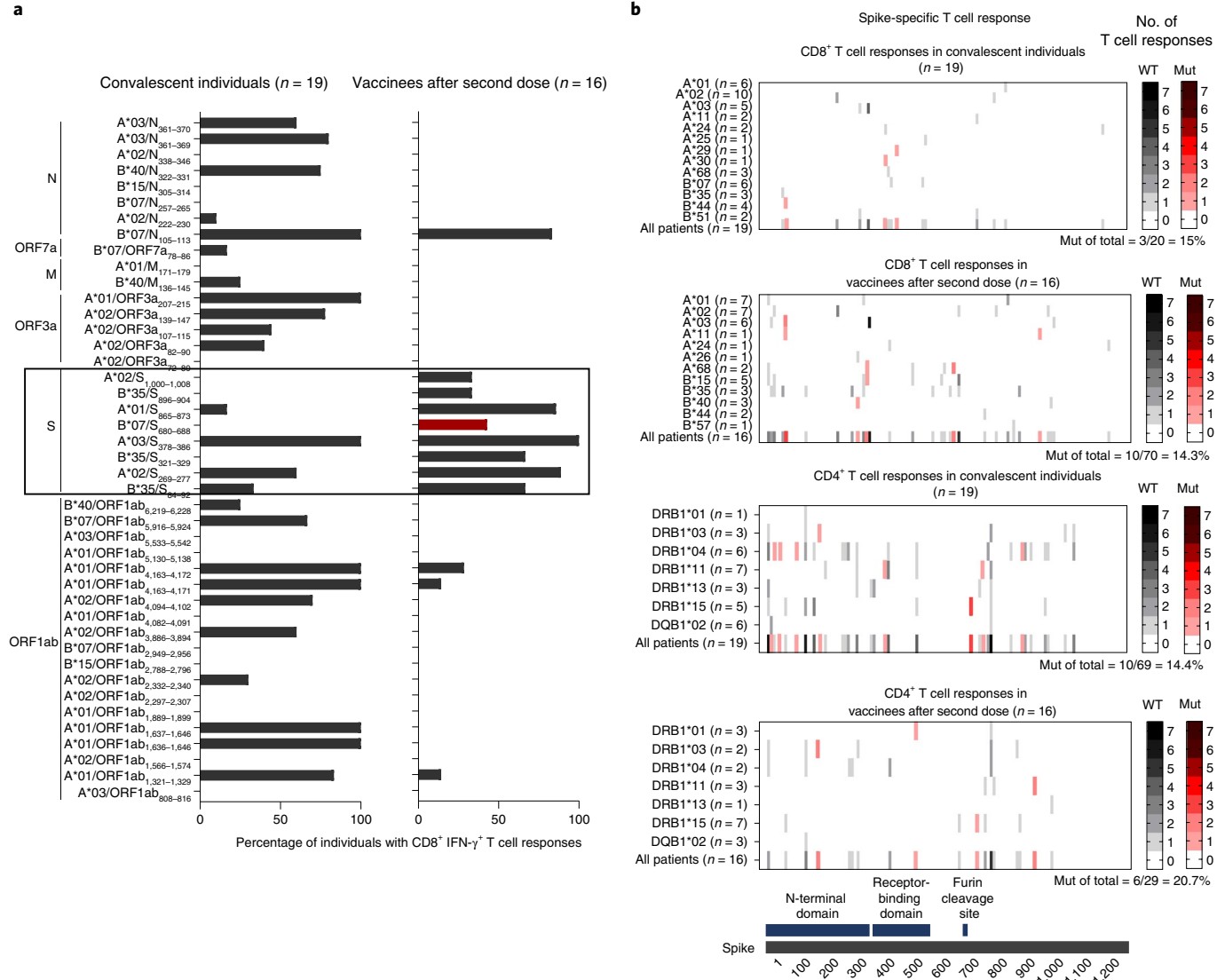

**Fig. 1 | CD8+ and CD4+ T-cell responses targeting conserved and mutated epitopes in Omicron/B.1.1.529, BA.1. a**, Percentages of CD8+ T-cell responses to previously described optimal CD8+ T-cell epitopes within the complete WT SARS-CoV-2 variant B proteome normalized to the HLA allotype. **b**, Number and location of spike-specific CD8+ (top) and CD4+ (bottom) T-cell responses to OLPs of the spike protein detectable in SARS-CoV-2 convalescent individuals and vaccinees after two doses of Pfizer/BioNTech mRNA vaccine are depicted. The heatmaps depict the compiled data of tested individuals. Epitopes with amino acid sequence variations in Omicron/B.1.1.529, BA.1 are marked in red. The number of tested individuals (per HLA allotype and in total) and percentage of total T-cell responses targeting variant epitopes are indicated. WT: epitope conserved between WT SARS-CoV-2 variant B and Omicron/B.1.1.529; Mut: epitope mutated in Omicron/B.1.1.529 compared to WT variant B SARS-CoV-2.

**Spike-specific T-cell epitope repertoire.** To analyse the spike-specific CD8+ T-cell response in convalescent individuals versus vaccinees in more detail, we assessed these responses using overlapping peptides (OLPs) spanning the whole spike protein. For all positive responses, we evaluated the OLP for the described optimal epitopes restricted by the HLA class I alleles expressed by the respective individual. If no matching optimal epitopes were previously described, we performed an in silico analysis to predict the most likely HLA class I restriction and optimal epitope. Using this comprehensive approach, we identified an overall substantially broader repertoire of spike-specific CD8+ T-cell responses in vaccinees (Fig. 1b (second panel) and Extended Data Fig. 1b (second panel)) compared to convalescent individuals (Fig. 1b (first panel) and Extended Data Fig. 1b (first panel)). Indeed, in convalescent individuals, no HLA class I allele restricted more than two spike-specific CD8+ T-cell epitopes, while several HLA class I alleles

restricted five or more spike-specific CD8+ T-cell epitopes in vaccinees. In addition, we detected more spike-specific CD8+ T-cell responses per individual in vaccinees compared to convalescent individuals (Extended Data Fig. 1c). Hence, the increased breadth of the spike-specific CD8+ T-cell response in vaccinees was evident at the individual and population levels.

We also analysed the CD4+ T-cell response using spike-spanning OLPs as described above. In contrast to the CD8+ T-cell response, the spike-specific CD4+ T-cell response showed a more limited repertoire of targeted epitopes after vaccination compared to infection (Fig. 1b, bottom). In particular, fewer spike-specific CD4+ T-cell epitopes were restricted by single HLA class II alleles (Fig. 1b, bottom) and fewer CD4+ T-cell responses were detectable per individual (Extended Data Fig. 1d) in vaccinees compared to convalescent individuals. Therefore, the spike-specific CD4+ T-cell repertoire was limited with regard to the individual- and population-based

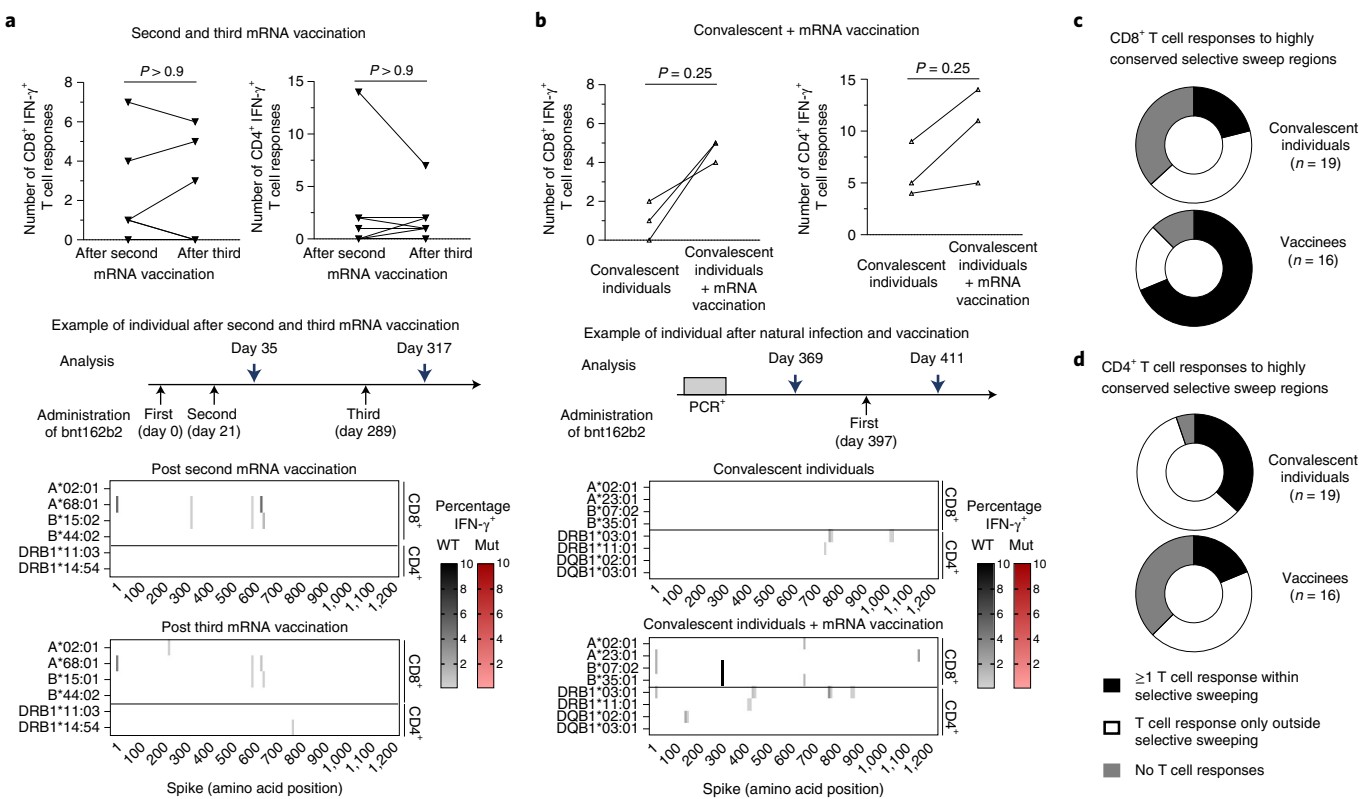

**Fig. 2 | Boosted vaccine- and infection-induced spike-specific CD8+ and CD4+ T-cell responses. a,b,** Number, location and percentages of spike-specific CD8+ and CD4+ T-cell responses to OLPs that are detectable in SARS-CoV-2 vaccinees after the second versus after the third dose of Pfizer/BioNTech mRNA vaccine (bnt162b2, measured 2–4 weeks after vaccination) (**a**) and in SARS-CoV-2 convalescent individuals who subsequently received a single dose of Pfizer/BioNTech mRNA boost vaccination (bnt162b2, measured 2 weeks after vaccination) (**b**). The heatmaps depict the data of one representative individual each. Targeted epitopes with sequence variations in Omicron/B.1.1.529, BA.1 are marked in red. **c,d,** Vaccinees and convalescent individuals with CD8+ (**c**) and CD4+ (**d**) T-cell responses within and outside highly conserved selective sweep regions in the spike protein are shown. Statistical analysis was performed with a two-sided Wilcoxon matched-pairs signed-rank test.

CD4+ T-cell response in vaccinees. Of note, the CD4+ and CD8+ spike-specific T-cell epitope repertoire was relatively stable over time in both vaccinees and convalescent individuals (Extended Data Fig. 2). Importantly, the fewer targeted spike-specific CD4+ T-cell epitopes in vaccinees exhibited high conservation between WT variant B and Omicron/B.1.1.529 SARS-CoV-2 (subvariants BA.1 and BA.2) as it is also the case for most targeted epitopes in convalescent individuals, similar to spike-specific CD8+ T-cell epitopes (Fig. 1b, variant epitopes in BA.1 in red, Extended Data Fig. 1b, variant epitopes in BA.2 in blue and Supplementary Table 3). While the differential responsiveness of CD4+ versus CD8+ T cells to peptide stimulation may limit comparison between the two T-cell subsets, mRNA vaccination appears to particularly broaden and thus increase a CD8+ T-cell response that targets conserved spike epitopes (Extended Data Fig. 1e).

**Booster effect on spike-specific T-cell epitope repertoire.** Next, to assess the effect of boosting vaccination- or infection-induced T-cell responses by mRNA vaccination on the spike-specific CD8+ T-cell repertoire, we again used overlapping spike peptides to map spike-specific CD8+ and CD4+ T-cell responses in longitudinally followed vaccinees getting their third vaccine dose (Pfizer/BioNTech mRNA vaccine; n = 7; Supplementary Table 1) and convalescent individuals who received an mRNA booster vaccination (n = 3; Supplementary Table 1). After the third mRNA vaccination, we observed a similarly broad and spike cross-recognizing CD8+ T-cell response and similarly limited but

still spike cross-recognizing CD4+ T-cell response compared to the completed initial immunization with two vaccine doses (Fig. 2a and Extended Data Fig. 3a). Interestingly, we detected CD4+ and CD8+ T-cell responses targeting more OLPs after the mRNA boost vaccination in convalescent individuals, representing a broader spike-specific T-cell repertoire (Fig. 2b and Extended Data Fig. 3b). Again, the identified CD8+ and CD4+ T-cell responses targeted epitopes that are conserved in Omicron/B.1.1.529. Thus, mRNA booster vaccination increased SARS-CoV-2-specific T-cell responses targeting conserved regions within the spike protein of Omicron/B.1.1.529 in convalescent individuals.

**Conservation of spike-specific T-cell epitopes.** To investigate whether the observed broader spike-specific CD8+ T-cell repertoire after mRNA vaccination may also be beneficial for potentially emerging future SARS-CoV-2 VOCs beyond Omicron/B.1.1.529, we analysed the T-cell response targeting highly conserved selective sweep regions in SARS-CoV-2 that were identified by Kang et al.[11] in convalescent individuals versus vaccinees. Selective sweep regions mediate per definition an evolutionary advantage; therefore, it is very likely that newly emerging SARS-CoV-2 VOCs also harbour high conservation within these regions. Four different selective sweep regions have so far been described in the spike protein of SARS-CoV-2 (ref. [11]) that also exhibit, as expected, a high degree of amino acid homology among the already evolved SARS-CoV-2 VOCs. For example, complete homology is present in VOC Delta, only one point mutation is present in VOC

Alpha or Beta, two mutations in VOC Gamma, seven point mutations in VOC Omicron/B.1.1.529, BA.1 and nine point mutations in VOC Omicron/B.1.1.529, BA.2 (Extended Data Fig. 4a,b). Importantly, compared to convalescent individuals more vaccinees showed spike-specific CD8+ T-cell responses targeting epitopes within the highly conserved selective sweep regions indicating a spike-specific CD8+ T-cell response with focused targeting of highly conserved regions after vaccination (Fig. 2c). A similarly focused spike-specific CD4+ T-cell response was not evident after vaccination (Fig. 2d). Hence, a broadly spike cross-recognizing CD8+ T-cell response is induced after mRNA vaccination that may be also reactive towards emerging SARS-CoV-2 VOCs in the future beyond Omicron/B.1.1.529.

## Discussion

In conclusion, our data indicate that (1) convalescent individuals target a variety of SARS-CoV-2-specific CD8+ T-cell epitopes over the complete SARS-CoV-2 proteome with spike-specific CD8+ T-cell responses being non-dominant; (2) in contrast to the CD4+ T-cell response, CD8+ T-cell responses in vaccinees are focused on a broader repertoire of highly conserved spike-specific CD8+ T-cell epitopes leading to an increased cross-recognizing potential; (3) boosting convalescent individuals with mRNA vaccination results in a broader spike-specific CD8+ T-cell response; and (4) CD8+ and CD4+ T-cell responses in both convalescent individuals and vaccinees target epitopes that are highly conserved between WT SARS-CoV-2 variant B, Omicron/B.1.1.529 and potentially future emerging SARS-CoV-2 variants and thus cross-recognize these variants. Hence, our data emphasize the relevance of mRNA vaccine-induced spike-specific CD8+ T-cell responses in combating emerging SARS-CoV-2 VOCs including Omicron/B.1.1.529 and support the benefit of also boosting convalescent individuals with mRNA vaccines.

## Methods

**Ethics.** Patients were recruited at the Freiburg University Medical Center between August 2019 and January 2022. Written informed consent was obtained from all participants. The study was conducted according to federal guidelines and local ethics committee regulations and the Declaration of Helsinki (first revision). The study was approved by the ethics committee of the University of Freiburg (nos. 21-1135 and 21-1372).

**Study cohort and clinical definitions.** Nineteen convalescent individuals after a mild course of COVID-19 were analysed (Supplementary Table 1). All patients were confirmed to have tested positive for SARS-CoV-2 using PCR with reverse transcription from an upper respiratory tract (nose and throat) swab tested at an accredited laboratory. The degree of COVID-19 severity was identified according to recommendations from the World Health Organization. Moreover, 16 individuals (all testing negative for anti-N-IgGs, (Mikrogen)) were screened 2–4 weeks after the second dose of mRNA vaccination (Pfizer/BioNTech BNT162) and 7 of the same individuals 2–4 weeks after the third mRNA vaccination (Pfizer/BioNTech BNT162). Three additional individuals were analysed who had a mild course of COVID-19 and were vaccinated once with the mRNA vaccine (Pfizer/BioNTech BNT162). The median age of vaccinated donors ($n = 16$) was 36 years; the median age of donors with a history of natural SARS-CoV-2 infection ($n = 19$) was also 36 years. The sex ratio of vaccinated donors was M/F 10/6; in donors with a history of natural SARS-CoV-2 infection, it was M/F 11/8. Participants did not receive any compensation for participating in the study.

**Peripheral blood mononuclear cell isolation.** Peripheral blood mononuclear cells (PBMCs) were isolated from blood samples anticoagulated with density gradient centrifugation (Pancoll Separation Medium; PAN Biotech) and subsequently stored at −80°C. Frozen PBMCs were thawed in Roswell Park Memorial Institute (RPMI) 1640 medium supplemented with 10% fetal calf serum, 1% penicillin/streptomycin and 1.5% HEPES buffer 1 mol l⁻¹ (complete medium; all additives from Thermo Fisher Scientific) until further usage.

**Peptides.** A total of 180 OLPs spanning the SARS-CoV-2 spike sequence (GenBank accession no. MN908947.3) were synthesized as 18-mer sliding by 7 amino acids and thus overlapping by 11 amino acids with a free amine NH₂ terminus and a free acid COOH terminus with standard Fmoc chemistry and a purity of >70% (Genaxxon Bioscience). Since two OLPs contained the two

amino acid residues with sequence modification in the 'stabilized' mRNA spike vaccine (K986P, V987P), these peptide variants were also synthesized and used, resulting in a total of 182 OLPs. In addition, 60 predescribed SARS-CoV-2-specific optimal CD8+ T-cell epitopes were synthesized. In the figures and Supplementary Table 2, we display only those 43 optimal epitopes that were tested in at least 3 HLA-matched individuals of each cohort (convalescent and vaccinated).

**In vitro expansion and intracellular IFN-γ staining using overlapping peptides or predescribed optimal CD8+ T-cell epitopes.** In vitro expansion with OLPs or optimal epitopes was performed as follows: 20% of the PBMCs were stimulated with a pool of all 182 SARS-CoV-2 spike OLPs or all 60 optimal epitopes (10 μg ml⁻¹) for 1 h at 37°C, washed and cocultured with the remaining PBMCs in RPMI medium supplemented with 20 U ml⁻¹ recombinant interleukin-2 (IL-2). On day 10, intracellular IFN-γ staining was performed with pooled OLPs (45 pools with 4 OLPs each). Therefore, cells were restimulated with OLP pools (50 μM), dimethyl sulfoxide as negative control or phorbol 12-myristate 13-acetate and ionomycin as positive control in the presence of brefeldin A and IL-2. After 5 h of incubation at 37°C, cells were stained for surface markers (CD8+, CD4+; Via-Probe) and intracellular markers (IFN-γ). Subsequently, on days 12–14 single OLPs of positive pools and HLA-matched optimal CD8+ T-cell epitopes were tested by intracellular cytokine staining. Viral amino acid sequences of positive individual OLPs were analysed for predescribed minimal epitopes[3,5,6,12–14] or the best HLA-matched predicted candidate using the Immune Epitope Database (IEDB, https://www.iedb.org/; we used two prediction algorithms, ANN 4.0 and NetMHCpan EL 4.123, for 8-mer, 9-mer and 10-mer peptides with a half maximal inhibitory concentration of <500 nM). The major histocompatibility complex class I (MHC class I) binding predictions were made using the IEDB analysis resource ANN aka NetMHC v4.0 tool or the IEDB analysis resource NetMHCpan v.4.0 tool. The MHC class II binding predictions were made using the IEDB recommended 2.22 analysis resource consensus tool (smm/nn/sturniolo).

**Multiparametric flow cytometry.** The following antibodies were used for the flow cytometry analysis: anti-CD8-APC (1:200, SK-1 clone); anti-CD4 eFluor 450 (1:250, RPA-T4 clone); anti-IFN-γ-FITC (1:8, 25723.11 clone); and fixable viability dye (1:200, 1:400, eFluor 506 clone). After cell fixation 2% paraformaldehyde/PBS (Sigma-Aldrich); acquisition was performed on a FACSCanto system (BD Biosciences). Data were collected with the FACSDiva software v.10.6.2 (BD Biosciences) and analysed with FlowJo v.10.0.7r2 (FlowJo LLC). The gating strategy is depicted in Extended Data Fig. 5.

**Enzyme-linked immunosorbent assay.** Spike-binding antibodies were assessed by anti-SARS-CoV-2 QuantiVac ELISA (IgG) (EUROIMMUN) detecting S1 IgG (<25.6 BAU ml⁻¹: negative; 25.6-35.1 BAU ml⁻¹: marginally positive; ≥35.2 BAU ml⁻¹: positive) according to the manufacturer's instructions. The SparkControl Magellan software v.2.2 was used for data collection.

**Sequence alignment.** Sequence homology analyses were performed in Geneious v.11.0.5 (https://www.geneious.com/) using Clustal Omega v.1.2.2 alignment with default settings. The reference genome of human SARS-CoV-2 (MN908947.3; Wuhan-Hu-1 isolate, WT variant B) was downloaded from the National Center for Biotechnology Information database (https://www.ncbi.nlm.nih.gov/). SARS-CoV-2 epitopes were then mapped to the corresponding protein alignment. SARS-CoV-2 VOCs (and subvariants) were identified via CoVariants (https://covariants.org/). Selective sweep regions were marked as described by Kang et al.[11]. Briefly, Kang et al. analysed a total of 136,114 complete SARS-CoV-2 genomes from the human host. Subsequently to alignment with MAFFT version 7, sweep regions were detected by using OmegaPlus and RAiSD, with the original SARS-CoV-2 isolate Wuhan-Hu-1 genome (NC_045512.2) as an outgroup. Common outliers were manually grouped into 8 regions of at least 50 base pairs; 4 of these were located in open reading frame 1ab (ORF1ab) and 4 in the S region. Selective sweep 1 $S_{323-434}$, selective sweep 2 $S_{524-545}$, selective sweep 3 $S_{888-919}$ and selective sweep 4 $S_{965-1,050}$ were identified by this method[11]. The SARS-CoV-2 VOC amino acid sequences were aligned to selective sweep regions and peptides were mapped to the spike protein to identify peptides that localize to the selective sweep regions.

**Statistics.** Statistical analysis was performed with Prism 9 (GraphPad Software). Statistical significance was assessed by two-tailed Mann–Whitney $U$-test, two-sided Wilcoxon matched-pairs signed-rank test and Spearman correlation; *$P < 0.05$, **$P < 0.01$, ***$P < 0.001$, ****$P < 0.0001$.

**Reporting Summary.** Further information on research design is available in the Nature Research Reporting Summary linked to this article.

## Data availability

Sequences of the tested epitopes, sequences of the spike overlapping peptides used as well as a list summarizing all CD8+ and CD4+ T-cell responses to overlapping peptides are available at the community repository Open Science Framework (https://www.cos.io/products/osf) and can be found via https://osf.io/zbk6q/. All requests for additional raw and analysed data and materials will be promptly

reviewed by the University of Freiburg Center for Technology Transfer to verify if the request is subject to any intellectual property or confidentiality obligations. Patient-related data not included in the paper were generated as part of clinical examination and may be subject to patient confidentiality. Any data and materials that can be shared will be released via a material transfer agreement. Source data are provided with this paper.

## Code availability
No custom code was generated or applied in this study.

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

## Acknowledgements
We thank all donors for participating in the current study and the FREEZE-biobank-Center for Biobanking of the Freiburg University Medical Center and Medical Faculty for support. This study was supported by grants from the German Federal Ministry of Education and Research (no. 01KI2077 to G.K., M.S., M.H. and R.T.) and the German Research Foundation (no. 272983813 to B.B., T.B., R.T., M.H. and C.N.-H.; grant no. 256073931 to B.B., M.S., R.T., M.H. and C.N.-H.; and grant no. 413517907 to H.L. and J.L.-M.). This work was also supported by the project 'Virological and immunological determinants of COVID-19 pathogenesis—lessons to get prepared for future pandemics (no. KA1-Co-02 'COVIPA')' and a grant from the Helmholtz Association Initiative and Networking Fund (to R.T. and M.H.). M.H. is also supported by the Margarete von Wrangell Fellowship (State of Baden-Württemberg).

## Author contributions
J.L.-M., H.L., K.W., V.K. and V.O. planned, performed and analysed the experiments with help from E.S.A., A.G., D.B.R., D.H. and M.R. H.L., N.R., V.G., D.A., S.R., T.W., D.S. and C.F.W. were responsible for donor recruitment. F.E. performed the four-digit HLA typing by next-generation sequencing. S.G., K.C., M.S. and G.K. provided virological expertise. B.B. and T.B. contributed to data interpretation. T.B. supervised the CD4[+] T-cell analysis and interpreted the data. R.T., M.H. and C.N.-H. designed the study and contributed to experimental design and planning. J.L.-M., H.L., V.O., R.T., M.H. and C.N.-H. interpreted the data and wrote the manuscript. T.B., C.N.-H., M.H. and R.T. are the shared last authors.

## Competing interests
The authors declare no competing interests.

## Additional information
**Extended data** is available for this paper at https://doi.org/10.1038/s41564-022-01106-y.

**Correspondence and requests for materials** should be addressed to Robert Thimme, Maike Hofmann or Christoph Neumann-Haefelin.

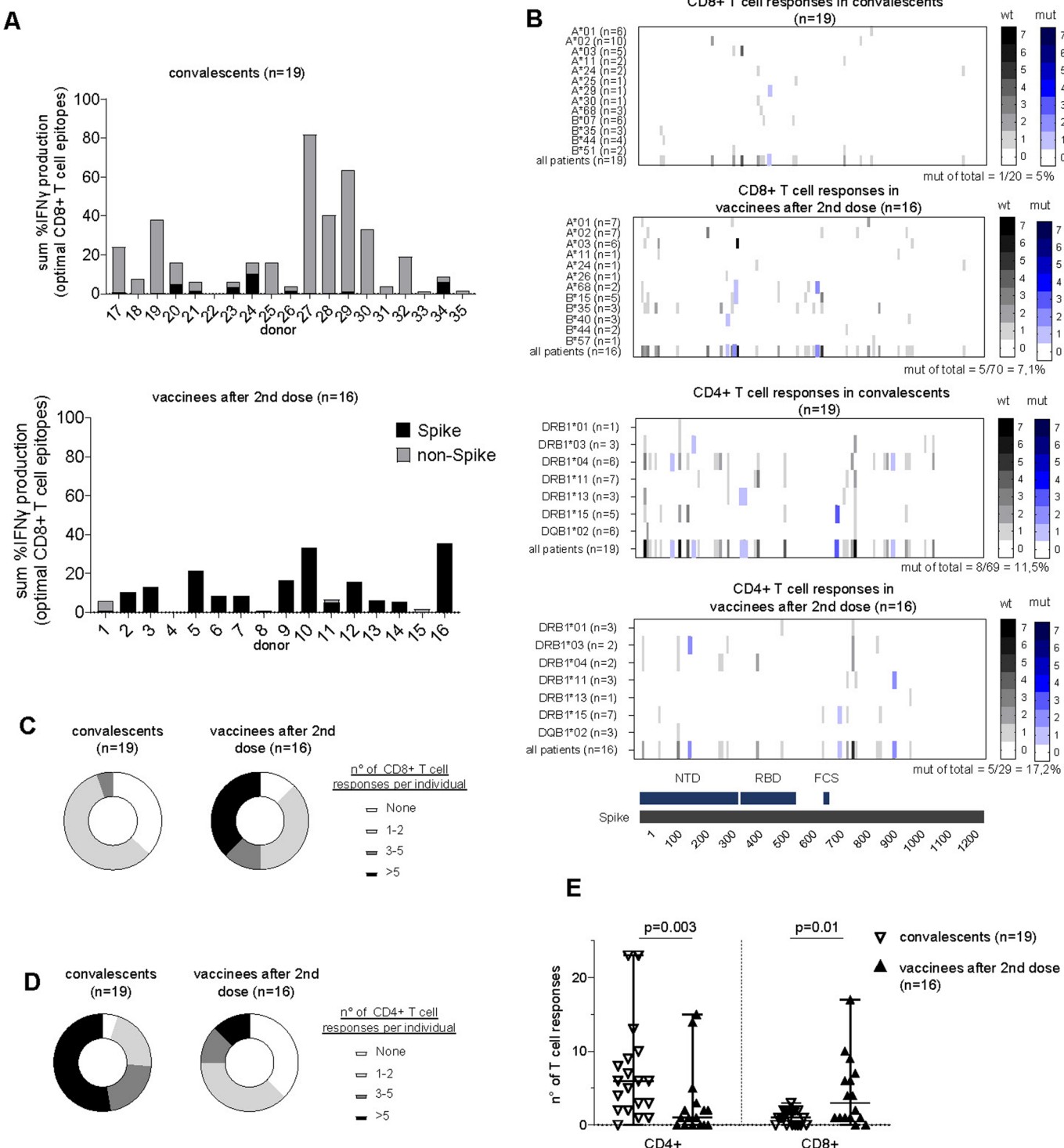

**Extended Data Fig. 1 | CD8 + and CD4 + T cell responses targeting spike-specific epitopes.** (**a**) Intensity (sum interferon-γ production of all CD8 + T cell responses per patient) to optimal CD8 + T cell epitopes. (**b**) Number and location of spike-specific CD8 + and CD4 + T cell responses to overlapping peptides (OLP) in SARS-CoV-2 convalescents and vaccinees after two doses of Pfizer/BioNTech mRNA vaccine. Epitopes with amino acid sequence variations in omicron BA.2 are marked in blue. Number of CD8 + (**c,e**) and CD4 + (**d,e**) T cell responses targeting spike-specific epitopes per individual in convalescents (n = 19) and vaccinees after two doses of Pfizer/BioNTech mRNA vaccine (n = 16). Numbers of tested individuals (per HLA allotype and in total) and % of total T cell responses targeting variant epitopes are indicated. Median and range are depicted, statistical analysis was performed with two-sided Mann-Whitney-Test. NTD:N-Terminal Domain; RBD: receptor-binding domain; FCS: furin cleavage site; wt: epitope conserved between ancestral and omicron SARS-CoV-2; mut: epitope mutated in omicron compared to ancestral SARS-CoV-2.

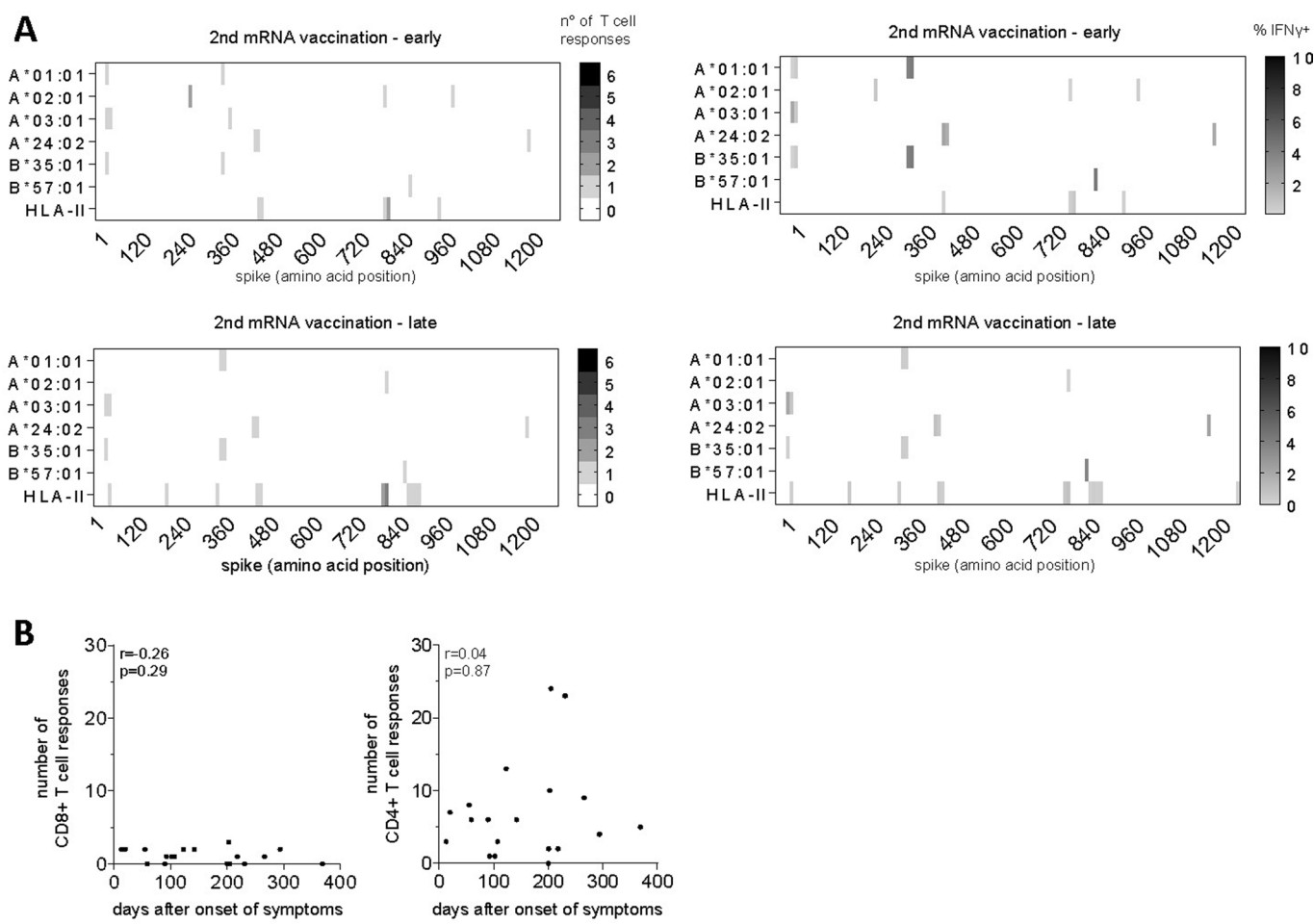

**Extended Data Fig. 2 | Stable T cell repertoire after infection and mRNA vaccination. (a)** Heatmaps showing percentage and location of CD4 + and CD8 + T cell responses targeting spike OLPs in three individuals at two different timepoints after mRNA vaccination. **(b)** Number of spike-specific CD4 + and CD8 + T cell responses to OLPs in correlation to days post symptoms onset in the first year after SARS-CoV-2 infection in the 19 convalescents. Spearman correlation analysis is depicted, p values are two-sided.

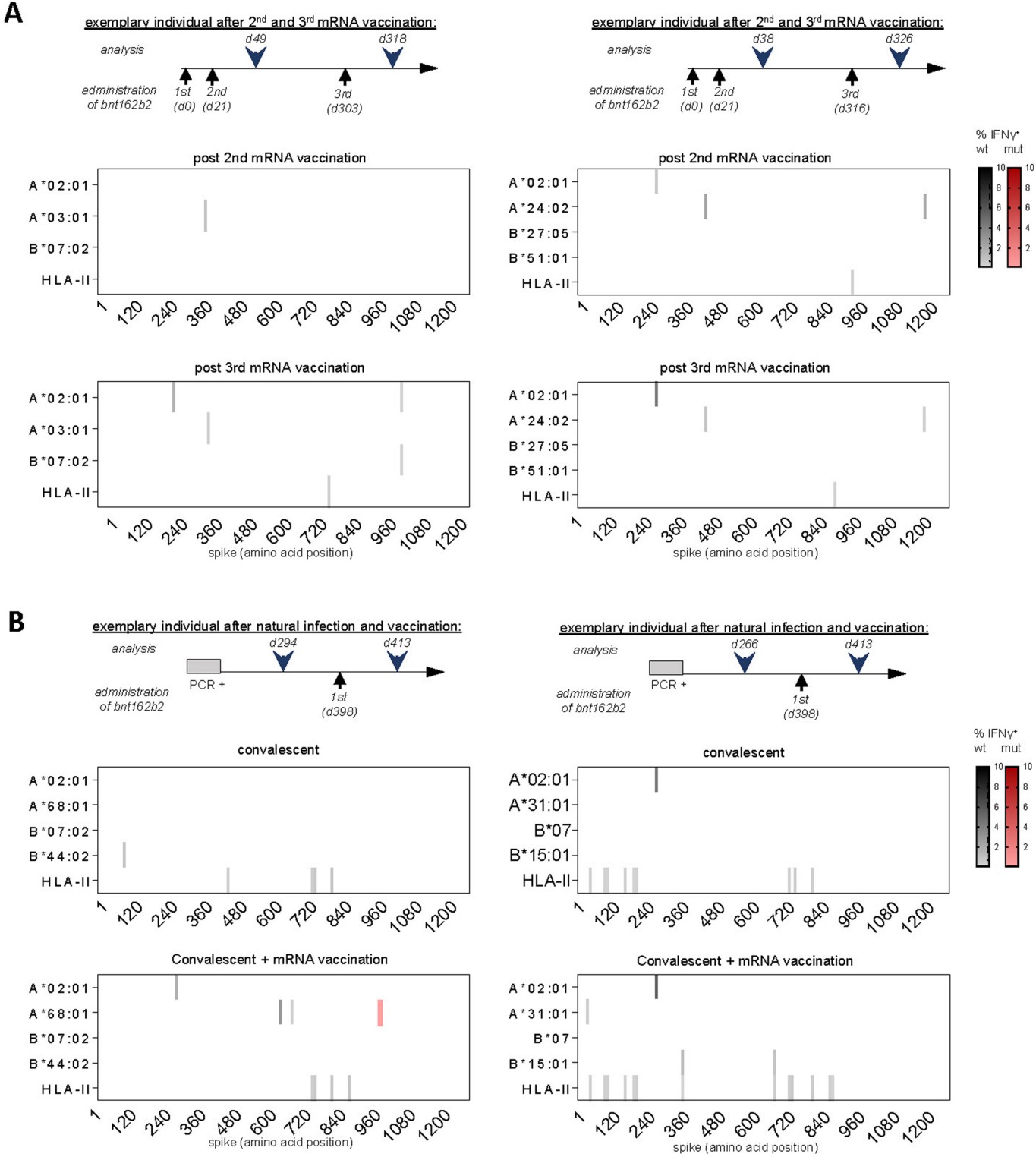

**Extended Data Fig. 3 | Boosted vaccine- and infection-induced spike-specific CD8 + and CD4 + T cell responses in exemplary individuals.** Number, magnitude and location of spike-specific CD8 + and CD4 + T cell responses to overlapping peptides (OLP) that are detectable in (**a**) exemplary SARS-CoV-2 vaccinees after the 2nd versus after the 3rd dose of Pfizer/BioNTech mRNA vaccine (bnt162b2) and in (**b**) exemplary SARS-CoV-2 convalescents who subsequently received a single dose of Pfizer/BioNTech mRNA boost vaccination (bnt162b2) are depicted. Epitopes with amino acid sequence variations in omicron BA.1 are marked in red.

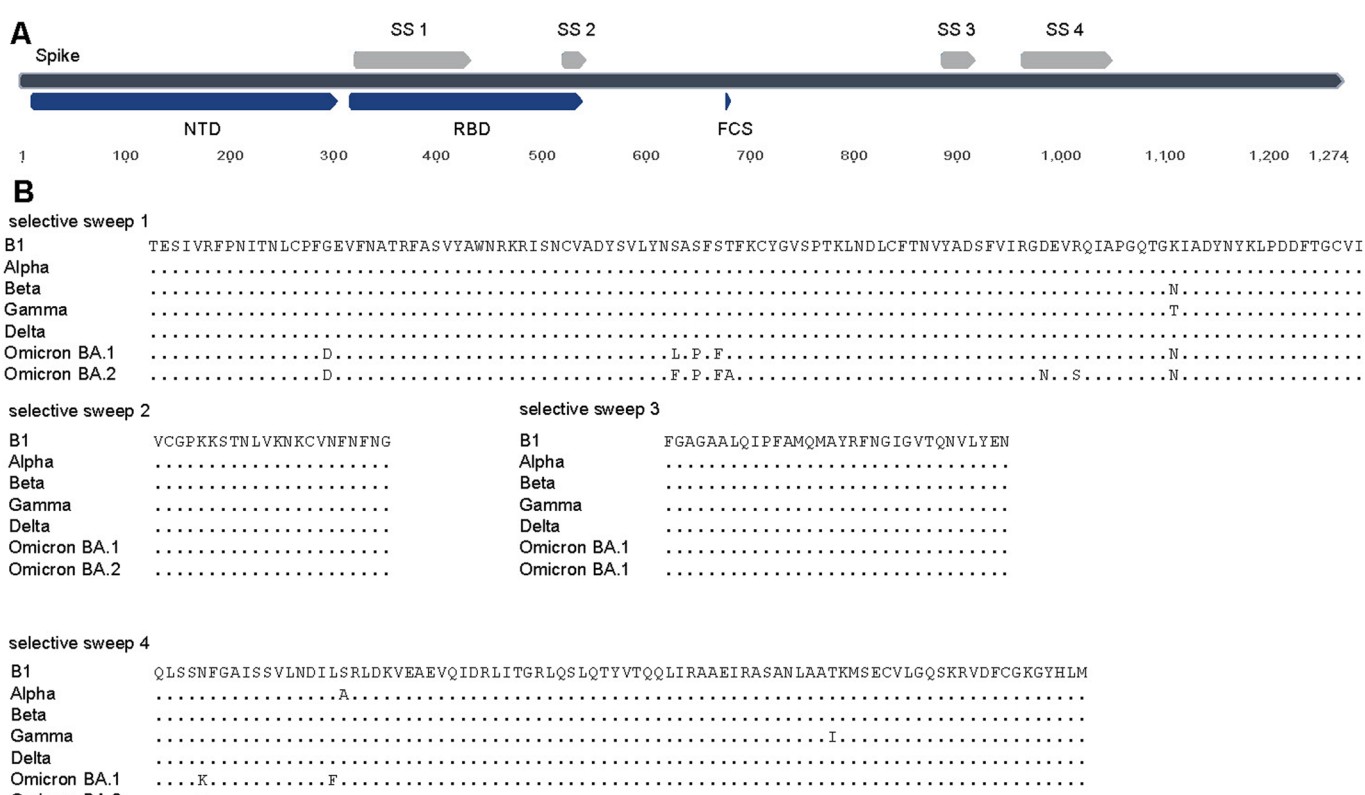

**Extended Data Fig. 4 | Selective sweep regions.** (**a**) Schematic representation of four previously identified selective sweep regions (SS1-4) within the spike protein of SARS-CoV-2. (**b**) Amino acid sequences of the selective sweep regions 1-4 in the spike protein of ancestral SARS-CoV-2 and VOC alpha, beta, gamma, delta and omicron (including subvariant BA.1 and BA.2).

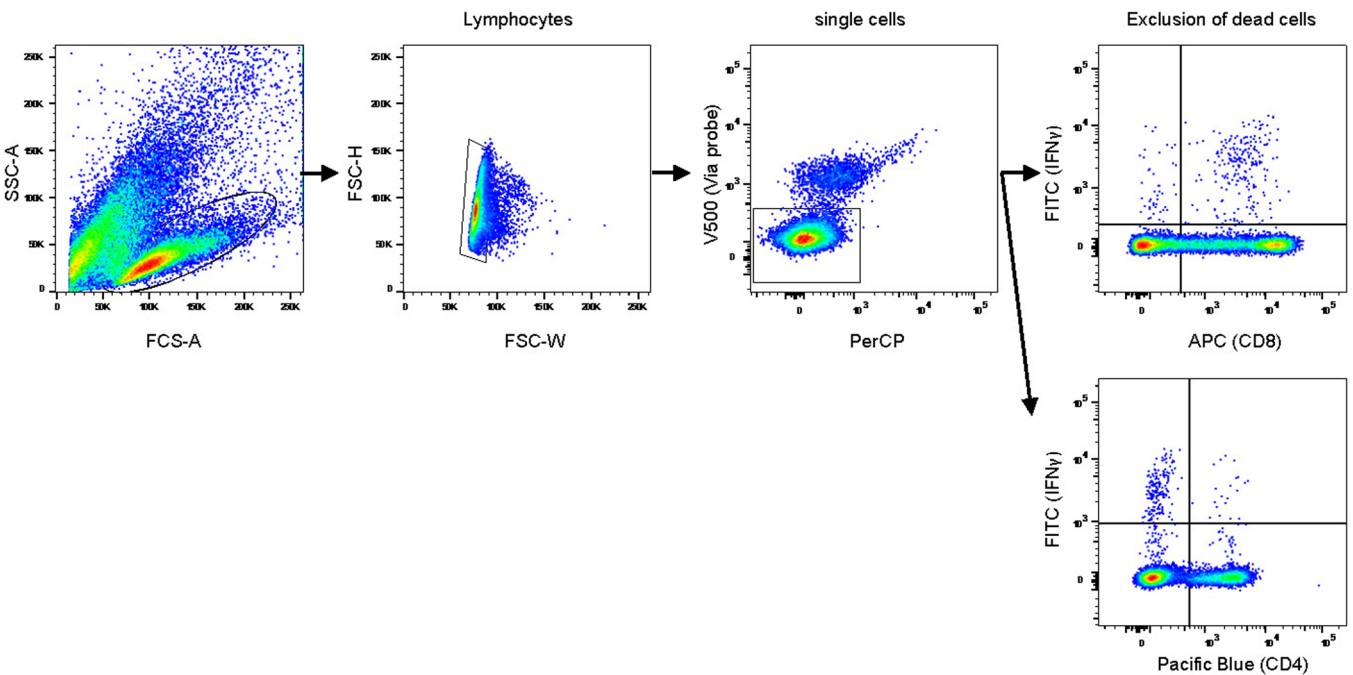

**Extended Data Fig. 5 | Gating strategy.** Lymphocytes were gated on FSC-A and SSC-A, Doublet exclusion on FSC-H and FSC-W, Exclusion of dead cells, Gating on CD8 + or CD4 + cells.

# nature research

# Reporting Summary

Nature Research wishes to improve the reproducibility of the work that we publish. This form provides structure for consistency and transparency in reporting. For further information on Nature Research policies, see our Editorial Policies and the Editorial Policy Checklist.

## Statistics

For all statistical analyses, confirm that the following items are present in the figure legend, table legend, main text, or Methods section.

| n/a | Confirmed | |
|---|---|---|
| ☐ | ☒ | The exact sample size (*n*) for each experimental group/condition, given as a discrete number and unit of measurement |
| ☐ | ☒ | A statement on whether measurements were taken from distinct samples or whether the same sample was measured repeatedly |
| ☐ | ☒ | The statistical test(s) used AND whether they are one- or two-sided *Only common tests should be described solely by name; describe more complex techniques in the Methods section.* |
| ☐ | ☒ | A description of all covariates tested |
| ☐ | ☒ | A description of any assumptions or corrections, such as tests of normality and adjustment for multiple comparisons |
| ☐ | ☒ | A full description of the statistical parameters including central tendency (e.g. means) or other basic estimates (e.g. regression coefficient) AND variation (e.g. standard deviation) or associated estimates of uncertainty (e.g. confidence intervals) |
| ☐ | ☒ | For null hypothesis testing, the test statistic (e.g. *F*, *t*, *r*) with confidence intervals, effect sizes, degrees of freedom and *P* value noted *Give P values as exact values whenever suitable.* |
| ☒ | ☐ | For Bayesian analysis, information on the choice of priors and Markov chain Monte Carlo settings |
| ☐ | ☒ | For hierarchical and complex designs, identification of the appropriate level for tests and full reporting of outcomes |
| ☒ | ☐ | Estimates of effect sizes (e.g. Cohen's *d*, Pearson's *r*), indicating how they were calculated |

*Our web collection on statistics for biologists contains articles on many of the points above.*

## Software and code

Policy information about availability of computer code

| Data collection | All software used to perform data collection are described in the methods section of the manuscript or the supportive information. Multiparametric Flow cytometry data was collected by FACSDiva software version 10.6.2 (BD, Germany). ELISA data was collected by SparkControl magellan software version 2.2. |
|---|---|
| Data analysis | Multiparametric Flow cytometry data was analyzed using FlowJo software version 10.6.2 (Treestar, Becton Dickinson). Visualization and statistical analysis was performed using GraphPad 8 software. Sequence homology analyses were performed in Geneious Prime 2020.0.3 (https://www.geneious.com/) using Clustal Omega 1.2.2 alignment with default settings. |

For manuscripts utilizing custom algorithms or software that are central to the research but not yet described in published literature, software must be made available to editors and reviewers. We strongly encourage code deposition in a community repository (e.g. GitHub). See the Nature Research guidelines for submitting code & software for further information.

## Data

Policy information about availability of data

All manuscripts must include a data availability statement. This statement should provide the following information, where applicable:
- Accession codes, unique identifiers, or web links for publicly available datasets
- A list of figures that have associated raw data
- A description of any restrictions on data availability

Sequences of tested epitopes, sequences of used spike overlapping peptides as well as a list summarizing all CD8+ and CD4+ T cell responses to overlapping peptides are available in the community repository "Open Science Framework" and can be found via https://osf.io/zbk6q/. Additional raw and analyzed data and materials include pseudonymized patient data that may be subject to confidentiality. Requests of this data are promptly reviewed by the University of Freiburg Center for Technology Transfer to verify if the request is subject to any intellectual property or confidentiality obligations. Patient-related data not included in the

paper were generated as part of clinical examination and may be subject to patient confidentiality. Any data and materials that can be shared will be released via a Material Transfer Agreement.

Materials and Correspondence:
Christoph Neumann-Haefelin: christoph.neumann-haefelin@uniklinik-freiburg.de
Maike Hofmann: maike.hofmann@uniklinik-freiburg.de
Robert Thimme: robert.thimme@uniklinik-freiburg.de
Hugstetter Straße 55, 79106 Freiburg, Germany

# Field-specific reporting

Please select the one below that is the best fit for your research. If you are not sure, read the appropriate sections before making your selection.

☒ Life sciences        ☐ Behavioural & social sciences        ☐ Ecological, evolutionary & environmental sciences

For a reference copy of the document with all sections, see nature.com/documents/nr-reporting-summary-flat.pdf

# Life sciences study design

All studies must disclose on these points even when the disclosure is negative.

| | |
|---|---|
| Sample size | Patients were recruited and  patient material was banked at the University Hospital Freiburg; inclusion criteria were: (1) 16 individuals that received a prime and boost vaccination with the mRNA vaccine bnt162b2/Comirnaty, (2) 19 acutely infected and convalescent individuals following a mild course of SARS-CoV-2 infection, SARS-CoV-2 infection was confirmed by positive PCR testing from oropharyngeal swab and/or SARS-CoV-2 spike IgG positive antibody testing. No sample size calculations were performed. 16 vaccinated health care workers gave informed consent and were available to donate blood samples. Therefore, similar numbers of COVID-19 convalescents were selected. |
| Data exclusions | No data was excluded from the analysis |
| Replication | Analyses were performed in independent experiments. Findings were reproducible. Flow cytometry analysis: 19 convalescent individuals following a mild course of SARS-CoV-2 infection were analyzed up to 7 months after infection (3 outliers up to 12 months). Moreover, 16 individuals were screened 2-4 weeks after first mRNA boost vaccination (Pfizer/BioNTech, bnt162b2) and 7 of the same individuals 2-4 weeks after second boost vaccination (Pfizer/BioNTech, bnt162b2) Three individuals were analyzed who had a mild course of SARS-CoV-2 infection and were vaccinated once with mRNA vaccine (Pfizer/BioNTech, bnt162b2). |
| Randomization | Vaccinated donors and donors with a history of natural SARS-CoV-2 infection were selected based on availability and HLA-typing. The covariates age and gender are well-documented: Median age of vaccinated donors was 36 years, donors with a history of natural SARS-CoV-2 infection was 36 years. The gender ratio of vaccinated donors was m/f: 10/6, donors with a history of natural SARS-CoV-2 infection was m/f: 11/8. |
| Blinding | Non-objective parameters were not included in the study design and standardized analyses were applied. Thus, blinding was not considered necessary since biased analysis can be excluded. |

# Reporting for specific materials, systems and methods

We require information from authors about some types of materials, experimental systems and methods used in many studies. Here, indicate whether each material, system or method listed is relevant to your study. If you are not sure if a list item applies to your research, read the appropriate section before selecting a response.

## Materials & experimental systems

| n/a | Involved in the study |
|---|---|
| ☐ | ☒ Antibodies |
| ☒ | ☐ Eukaryotic cell lines |
| ☒ | ☐ Palaeontology and archaeology |
| ☒ | ☐ Animals and other organisms |
| ☐ | ☒ Human research participants |
| ☒ | ☐ Clinical data |
| ☒ | ☐ Dual use research of concern |

## Methods

| n/a | Involved in the study |
|---|---|
| ☒ | ☐ ChIP-seq |
| ☐ | ☒ Flow cytometry |
| ☒ | ☐ MRI-based neuroimaging |

# Antibodies

| | |
|---|---|
| Antibodies used | Anti-CD8-APC (SK-1, 1:200), BD, Cat#345775<br>anti-CD4-efluor450 (RPA-T4, 1:250), eBioscience, Cat#48-0049<br>anti-IFN-γ-FITC (25723.11, 1:8), BD, Cat#340449<br>fixable Viability Dye (eFluor506 1:200, 1:400), eBioscience, Cat#65-0866 |

| Validation | All antibodies were obtained from commercial cendors and we based specificity on descriptions and information provided in corresponding data sheets available and provided by the manufacturers. Standardized analysis in different cohorts, antibody titration on PBMCs including unstained controls, comparisons of different antibody clones and conjugates and validated by publications:<br>CD4: antibody titration on PBMCs; control clones SK3; using B cells as negative control<br>CD8: antibody titration on PBMCs; control clone GHI/75; using B cells as negative control<br>IFNɣ, clone 25723.11: antibody titration on PBMCs; control clone 4S.B3; validated with respect to differential expression of activated and non-activated T cell subpopulations<br>Viability Dye was titrated on PBMCs; validated with respect to differential staining of live and dead cell populations |
|---|---|

# Human research participants

Policy information about <u>studies involving human research participants</u>

| Population characteristics | Median age of vaccinated donors (n=16) was 36 years, donors with a history of natural SARS-CoV-2 infection (n=19) was 36 years. The gender ratio of vaccinated donors was m/f: 10/6, donors with a history of natural SARS-CoV-2 infection was m/f: 11/8. |
|---|---|
| Recruitment | Vaccinated donors as well as SARS-CoV-2-infected and SARS-CoV-2-convalescent patients were recruited at the University Hospital Freiburg; self-selection bias or other biases can be excluded since several people were included in the recruitment. Samples were banked and retrospectively selected. Banked samples from sex-, age- and HLA-matched vaccinated and convalescent individuals were retrospectively selected. |
| Ethics oversight | Written informed consent was obtained from all participants and the study was conducted according to federal guidelines, local ethics committee regulations (Ethik-Kommission der Albert-Ludwigs-Universität Freiburg, Freiburg, Germany; vote #: 322/20, #21-1135 and 315/20) and the Declaration of Helsinki (1975). |

Note that full information on the approval of the study protocol must also be provided in the manuscript.

# Flow Cytometry

## Plots

Confirm that:

☒ The axis labels state the marker and fluorochrome used (e.g. CD4-FITC).

☒ The axis scales are clearly visible. Include numbers along axes only for bottom left plot of group (a 'group' is an analysis of identical markers).

☒ All plots are contour plots with outliers or pseudocolor plots.

☒ A numerical value for number of cells or percentage (with statistics) is provided.

## Methodology

| Sample preparation | Cryopreserved isolated human PBMCs were thawed and prepared for flow cytometry or in vitro expansion described in the methods section |
|---|---|
| Instrument | FACSCanto II |
| Software | FlowJo_v10.6.2 (Treestar) |
| Cell population abundance | Abundance of SARS-CoV-2-specific T cells are low (<10^-4 %) |
| Gating strategy | Lymphocytes gated on FSC-A and SSC-A, Doublet exclusion on FSC-H and FSC-W, Exclusion of dead cells, Gating on CD8+ or CD4+ cells. |

☒ Tick this box to confirm that a figure exemplifying the gating strategy is provided in the Supplementary Information.

