## [Peer Review File · Nature Microbiology]

Peer Review Information

Journal: Nature Microbiology

Manuscript Title: SARS-CoV-2-specific T cell epitope repertoire in convalescent and mRNA-vaccinated individuals

Corresponding author name(s): Maike Hofmann

Reviewer Comments & Decisions:Decision Letter, initial version:

Dear Dr Hofmann,

Thank you for your patience while your manuscript "SARS-CoV-2 specific T cells induced by both SARS-CoV-2 infection and mRNA vaccination broadly cross-recognize omicron" was under peer-review at Nature Microbiology. It has now been seen by 3 referees, whose expertise and comments you will find at the of this email. You will see from their comments below that while they find your work of interest, some important points are raised. We are very interested in the possibility of publishing your study in Nature Microbiology, but would like to consider your response to these concerns in the form of a revised manuscript before we make a final decision on publication.

In particular, you will see that reviewer #3 raises a concern about the experimental strength supporting the main conclusion of the paper that will need to be resolved. In addition, given that this topic is very time sensitive we will need your responses and revision back as fast as possible. The rest referees' reports are clear and the remaining issues should be straightforward to address.

If you have not done so already please begin to revise your manuscript so that it conforms to our Brief Communication format instructions at <http://www.nature.com/nmicrobiol/info/final-submission/>

Our normal length limit for a Brief Communication to Nature Microbiology with no more than 2 small display items (figures or tables) is 1,250 words. We have some flexibility, and can allow a revised manuscript at 1,500 words, but please consider this a firm upper limit.

Some reduction could be achieved by focusing any introductory material and moving it to the start of your opening 'bold' paragraph, whose function is to outline the background to your work, describe in a sentence your new observations, and explain your main conclusions. The discussion should also be limited. Methods should be described in a separate section following the discussion, we do not place a word limit on Methods.

Nature Microbiology titles should give a sense of the main new findings of a manuscript, and should not contain punctuation. Please keep in mind that we strongly discourage active verbs in titles, and that they should ideally fit within 90 characters each (including spaces).

Please include a data availability statement as a separate section after Methods but before references, under the heading "Data Availability". This section should inform readers about the availability of the data used to support the conclusions of your study. This information includes accession codes to public repositories (data banks for protein, DNA or RNA sequences, microarray, proteomics data etc...), references to source data published alongside the paper, unique identifiers such as URLs to data repository entries, or data set DOIs, and any other statement about data availability. At a minimum, you should include the following statement: "The data that support the findings of this study are available from the corresponding author upon request", mentioning any restrictions on availability. If DOIs are provided, we also strongly encourage including these in the Reference list (authors, title, publisher (repository name), identifier, year). For more guidance on how to write this section please

see:

<http://www.nature.com/authors/policies/data/data-availability-statements-data-citations.pdf>

To improve the accessibility of your paper to readers from other research areas, please pay particular attention to the wording of the paper's opening bold paragraph, which serves both as an introduction and as a brief, non-technical summary in about 150 words. If, however, you require one or two extra sentences to explain your work clearly, please include them even if the paragraph is over-length as a result. The opening paragraph should not contain references. Because scientists from other sub-disciplines will be interested in your results and their implications, it is important to explain essential but specialised terms concisely. We suggest you show your summary paragraph to colleagues in other fields to uncover any problematic concepts.

If your paper is accepted for publication, we will edit your display items electronically so they conform to our house style and will reproduce clearly in print. If necessary, we will re-size figures to fit single or double column width. If your figures contain several parts, the parts should form a neat rectangle when assembled. Choosing the right electronic format at this stage will speed up the processing of your paper and give the best possible results in print. We would like the figures to be supplied as vector files - EPS, PDF, AI or postscript (PS) file formats (not raster or bitmap files), preferably generated with vector-graphics software (Adobe Illustrator for example). Please try to ensure that all figures are non-flattened and fully editable. All images should be at least 300 dpi resolution (when figures are scaled to approximately the size that they are to be printed at) and in RGB colour format. Please do not submit Jpeg or flattened TIFF files. Please see also 'Guidelines for Electronic Submission of Figures' at the end of this letter for further detail.

Figure legends must provide a brief description of the figure and the symbols used, within 350 words, including definitions of any error bars employed in the figures.

Please include a statement before the acknowledgements naming the author to whom correspondence and requests for materials should be addressed.

Finally, we require authors to include a statement of their individual contributions to the paper -- such as experimental work, project planning, data analysis, etc. -- immediately after the acknowledgements. The statement should be short, and refer to authors by their initials. For details please see the Authorship section of our joint Editorial policies at http://www.nature.com/authors/editorial_policies/authorship.html

- * include a point-by-point response to any editorial suggestions and to our referees. Please include your response to the editorial suggestions in your cover letter, and please upload your response to the referees as a separate document.

- * ensure it complies with our format requirements for Letters as set out in our guide to authors at www.nature.com/nmicrobiol/info/gta/

- * state in a cover note the length of the text, methods and legends; the number of references; number and estimated final size of figures and tables

- * resubmit electronically if possible using the link below to access your home page:

{redacted}

- *This url links to your confidential homepage and associated information about manuscripts you may have submitted or be reviewing for us. If you wish to forward this e-mail to co-authors, please delete this link to your homepage first.

Please ensure that all correspondence is marked with your Nature Microbiology reference number in the subject line.

Nature Microbiology is committed to improving transparency in authorship. As part of our efforts in this direction, we are now requesting that all authors identified as 'corresponding author' on published papers create and link their Open Researcher and Contributor Identifier (ORCID) with their account on the Manuscript Tracking System (MTS), prior to acceptance. This applies to primary research papers only. ORCID helps the scientific community achieve unambiguous attribution of all scholarly contributions. You can create and link your ORCID from the home page of the MTS by clicking on 'Modify my Springer Nature account'. For more information please visit www.springernature.com/orcid.

We hope to receive your revised paper as fast as possible but maximum within three weeks. If you cannot send it within this time, please let us know.

{redacted}

Reviewer Expertise:

Referee #1:

Referee #2:

Referee #3:

Reviewers Comments:

Reviewer #1 (Remarks to the Author):

In the study SARS-CoV-2 specific T cells induced by both SARS-CoV-2 infection and mRNA vaccination broadly cross-recognize omicron, Lang-Meli et al. Importantly, they also compare responses induced by natural infection vs vaccination down to the epitope level. This is of importance as it is still unclear if different or same epitopes are induced by either mechanism. The authors further address the question if there are differences in naturally infected then vaccinated compared to boosted vaccinees. Comments to the authors:

- There is some concern as the authors for rely on previously identified CD8 T cells epitopes. It cannot be excluded that the authors would have identified novel CD8 epitopes in either vaccinated or naturally infection boosted vaccinees. Please add this limitation to the discussion.
- It is not entirely clear how these selective sweep regions have been defined and how these 4 have been selected. Please add more detail to the methods section.
- Magnitude of T cell responses was compared after 10 days of stimulation. CD4 and CD8 T cells have very different kinetics of stimulation in vitro. While this should not affect the epitope specificity the reviewer doesn't think the authors can comment on the magnitude differences as shown in Figure 1F. Please consider deleting this.

Reviewer #2 (Remarks to the Author):

In this manuscript, Lang-Meli and colleagues study the profile of T cells responding to both mRNA vaccination and previous infection. They demonstrate interesting changes in the specificities of responding T cells following either vaccination or infection and go one step further and study the responses following booster vaccination in both cases. They further demonstrate that these responses are able to cross-react to Omicron in many cases and that some are in conserved regions of spike that may therefore respond to any future arising variants. This is an interesting and well crafted study that

will likely benefit the field as a whole.

One comment is that it seems like the responses presented in figure 2 are much less broad than what is presented in figure 1 and this isn't discussed. In addition, it appears that booster vaccination reduces the breadth of the CD4 T cell response but this isn't discussed in the manuscript. Both of these points should be clarified in the text.

Reviewer #3 (Remarks to the Author):

Lang-Meli et al. describe in their manuscript SARS-CoV-2 T cell responses in convalescent and mRNA-vaccinated individuals using (1) a panel of predescribed SARS-CoV-2 T cell epitopes and (2) whole spike protein overlapping peptide pools. They report that CD8 T cell responses targeting different spike peptides are lower in convalescent donors compared to vaccinees. For CD4+ T cells it was shown vice versa. Whereas the authors clearly demonstrate the differences in the T cell repertoire between convalescent and vaccinees, the data does not support the conclusion made by the authors in the title of the manuscript. T cell responses were all analyzed against the ancestral SARS-CoV-2 sequence and not against the omicron sequence. The statement that SARS-CoV-2-specific T cells cross-recognize omicron is not supported by this data. Furthermore, the authors suggest that the only relevant T cell responses are those against the spike protein, which indeed is not the case. A broad T cell response targeting multiple epitopes was shown to be most relevant to combat COVID-19. Intensity in terms of % of SARS-CoV-2 specific T cells in the two population was not assessed at all, which might be of central relevance to assess the quality of T cell response.

Please find below the detailed comments on the manuscript and figures:

- Of the 19 convalescent donors three were also vaccinated. Did the authors include samples of those donors before vaccination? Otherwise, these three should not be included in the cohort of convalescents and only used for the longitudinal analysis.
- At which time point after vaccination or after infection were the blood samples collected? That should clearly be stated e.g. in Sup Table 1.
- The authors should provide a table with the 43 tested peptide sequences. In the method section, 60 peptides were described. Where does the difference between 43 and 60 come from?
- Regarding the overlapping peptides: line 105: 180 peptides, method section line 29: 182 peptides, method section line 38: 181 peptides. What is the correct number?
- Regarding the one spike-derived peptide affected by omicron (red in Figure 1A), a supplementary Figure or Table showing the sequence of the peptide and the sequence of the omicron variant would be helpful. Is it only affected by one amino acid? In which position? Anker position? The same could be analyzed for the affected peptides in Figure 1B and D. Why did the authors not test the convalescent and the vaccinated donors with the omicron peptides? Only by analyzing the T cell response against ancestral peptides AND omicron peptides can truly demonstrate if the T cells are able to recognize omicron sequences or if the T cell response in total is not reduced or affected by omicron.
- Figure 1: All the Figures should be labeled with "convalescents (n=19)" and "vaccinees (n = 16)". Do the authors show only vaccinees after the second dose? This should be clearly labeled within the figure and in the legend. Furthermore, the figure legend is not sufficient enough to understand the figure e.g. explanation of wt vs mut in B and D or abbreviations such as NTD, RBD, FCS are missing.
- Figure 1A: Are the frequency of donors who recognize a peptide normalized to the HLA allotypes? If not, why did 100% of convalescents recognize A*01-restricted peptides where only 6/19 convalescents are A*01 positive and why did 100% of the vaccinees recognize the A*03-restricted peptide where only 6/16 vaccinees are A*03 positive. How do the authors explain this?
- Figure 1B/D: Are the n numbers behind the HLA allotypes the number of donors with this allotype? One summary line per plot summarizing all donors independent of the allotype would be helpful.
- Figure 1F: For the convalescents CD4+ only 18 points are visible. That should be 19, shouldn't it?
- Regarding the statistics: A paired t test is not correct here since the donors of the different cohorts are not paired. I would suggest a nonparametric test such as the Mann-Whitney-Test.
- Did the authors test the donors of the vaccinee cohort for anti-nucleocapsid antibodies to be sure they did not have a previously unknown SARS-CoV-2 infection?
- The data in the extended data Figure 1 does not match the statement in the text (line 112/113). The authors discuss the differences between convalescent and vaccinees and not the longitudinal analysis within the different cohorts, which is shown in the figure.

- Extended Data 1B: The different donors in these plots should be clearly labeled. The number of T cell responses depends on the HLA type of the donors hampering a comparison over all donors in the same plot.
- Using the terms cross-reactive and spike-specific together is very confusing and mutually exclusive (line 132/140). Cross-reactive peptides can be recognized by non-SARS-CoV-2-specific common cold coronavirus-primed T cells. The authors did not know if the vaccinees have cross-reactive T cells targeting the spike peptides before the vaccination.
- Figure 2: Did the authors test the data for normality before using a t test? Maybe a nonparametric test such as Wilcoxon matched-pairs signed rank test would be better.
- The data about the sweep regions are not convincing since omicron showed 7 mutations in these regions.
- The extended data Figure 3 is not referred to in the text. For Pacific Blue the stained marker CD4 is not labeled on the axis.
- Some typos: line 1: SARS-CoV-2-specific T cells, line 71: breakthrough, Supplementary Table 1: A*30:04 for donor 35
- Supplementary Table 1: List of abbreviations is missing. Mild defines the course of COVID-19 not the natural infection.

Author Rebuttal to Initial comments

REFEREE #1:

In the study SARS-CoV-2 specific T cells induced by both SARS-CoV-2 infection and mRNA vaccination broadly cross-recognize omicron by Lang-Meli et al., importantly, they also compare responses induced by natural infection vs vaccination down to the epitope level. This is of importance as it is still unclear if different or same epitopes are induced by either mechanism. The authors further address the question if there are differences in naturally infected then vaccinated compared to boosted vaccinees.

We would like to thank this reviewer for the positive feedback and valuable comments regarding our study, especially with respect to acknowledging our comparative approach of identifying T cell responses down to the epitope level.

Comments to the authors:

- There is some concern as the authors rely on previously identified CD8 T cells epitopes. It cannot be excluded that the authors would have identified novel CD8 epitopes in either vaccinated or naturally infection boosted vaccinees. Please add this limitation to the discussion.

We agree with this reviewer that with respect to the targeted epitope repertoire outside the spike protein, we completely rely on previously identified CD8+ T cell epitopes and thus neglect novel epitopes. However, to overcome the limitation of using pre-described epitopes with respect to spike-specific T cell responses, we additionally performed stimulation with overlapping peptides, followed by single-peptide intracellular cytokine read-out and *in silico* prediction of HLA restriction and optimal epitopes. Indeed, by using this approach, we detected substantially more CD8+ T cell responses targeting novel spike-specific T cell epitopes in vaccinees compared to convalescent donors (Fig. 1B and Extended Data Fig. 1C+E).

- It is not entirely clear how these selective sweep regions have been defined and how these 4 have been selected. Please add more detail to the methods section.

We have now added more detailed information to the Methods section (page 8, lines 253-263) about how the selective sweep regions were defined. In particular, selective sweep regions were defined according to Kang *et al.* (Kang L, He G, Sharp AK, et al. A selective sweep in the Spike gene has driven SARS-CoV-2 human adaptation. Cell 2021;184:4392-400). In brief, Kang et al. analyzed a total of 136,114 complete SARS-CoV-2 genomes from the human host. Subsequently to alignment with MAFFT, sweep regions were detected by using OmegaPlus and RAiSD, with the original SARS-CoV-2 isolate Wuhan-Hu-1 genome (NC_045512.2) as an outgroup. The common outliers were manually grouped into eight region of at least 50 bp, four of these were located in ORF1ab and four in the S region. Selective sweep 1 S₃₂₃₋₄₃₄, selective sweep 2 S₅₂₄₋₅₄₅, selective sweep 3 S_{888-S₉₁₉}, selective sweep 4 S_{965-S_{1,050}} were identified by this method. VOC amino acid sequences were aligned to selective sweep regions; and peptides were mapped to the spike protein, to identify peptides that localize to the selective sweep regions.

- Magnitude of T cell responses was compared after 10 days of stimulation. CD4 and CD8 T cells have very different kinetics of stimulation *in vitro*. While this should not affect the epitope specificity the reviewer doesn't think the authors can comment on the magnitude differences as shown in Figure 1F. Please consider deleting this.

We appreciate the comment by the reviewer regarding previous Fig. 1F, now Extended Data Fig. 1E in the revised manuscript. We now mention the limited comparability of peptide-expansion of CD4+ versus CD8+ T cell responses on page 4, lines 113-115 of the revised manuscript. Since expansion procedures were, however, the same in vaccinees versus convalescents, the observed difference in the ratio between CD4+ and CD8+ T cell responses between these two groups may represent a biological relevant phenomenon.

REFEREE #2:

In this manuscript, Lang-Meli and colleagues study the profile of T cells responding to both

mRNA vaccination and previous infection. They demonstrate interesting changes in the specificities of responding T cells following either vaccination or infection and go one step further and study the responses following booster vaccination in both cases. They further

demonstrate that these responses are able to cross-react to Omicron in many cases and that some are in conserved regions of spike that may therefore respond to any future arising variants. This is an interesting and well-crafted study that will likely benefit the field as a whole.

We would like to thank this reviewer for acknowledging the strengths of our study and for the valuable remarks.

- One comment is that it seems like the responses presented in figure 2 are much less broad than what is presented in figure 1 and this isn't discussed.

In Fig. 1, the heatmaps depict a compilation of T cell responses from several donors whereas in Fig. 2, the heatmaps show the responses of one representative donor each. We have now better clarified this discrepancy in the respective figure legends. The median numbers of T cell responses per individual are, however, comparable (convalescents, CD4+: Fig. 1: 6, Fig. 2: 5; CD8+: Fig. 1: 1, Fig. 2: 1; vaccinees: CD4+: Fig. 1: 1, Fig. 2: 1; CD8+: Fig. 1: 3, Fig. 2: 1).

- In addition, it appears that booster vaccination reduces the breadth of the CD4 T cell response but this isn't discussed in the manuscript. Both of these points should be clarified in the text.

In two of seven donors the numbers of detectable CD4+ T cell responses were reduced after 3rd dose vaccination, in other two donors the numbers were increased and in three donors the numbers remained similar (Fig. 2A, right panel; $p>0.9$). Hence, from these data, it cannot be concluded that the breadth of the CD4+ T cell response is reduced after the 3rd dose vaccination. In the previous version of the manuscript, we indeed chose an 'exemplary' patient that did not well represent the overall results. We have now replaced the exemplary data by data from a more representative patient (Fig. 2A, lower part).

REFEREE #3:

Lang-Meli et al. describe in their manuscript SARS-CoV-2 T cell responses in convalescent and mRNA-vaccinated individuals using (1) a panel of pre-described SARS-CoV-2 T cell epitopes and (2) whole spike protein overlapping peptide pools. They report that CD8 T cell responses targeting different spike peptides are lower in convalescent donors compared to vaccinees. For CD4+ T cells it was shown vice versa. Whereas the authors clearly demonstrate the differences in the T cell repertoire between convalescent and vaccinees, the data does not support the conclusion made by the authors in the title of the manuscript. T cell responses were all analyzed against the ancestral SARS-CoV-2 sequence and not against the omicron sequence. The statement that SARS-CoV-2-specific T cells cross-recognize omicron is not supported by this data. Furthermore, the authors suggest that the only relevant T cell responses are those against the spike protein, which indeed is not the case. A broad T cell response targeting multiple epitopes was shown to be most relevant to combat COVID-19. Intensity in terms of % of SARS-CoV-2 specific T cells in the two population was not assessed at all, which might be of central relevance to assess the quality of T cell response.

We would like to thank this reviewer for very thoroughly reviewing our manuscript and for the valuable comments that have helped to clarify and improve our manuscript. Indeed, the focus of this study was to dissect to what extent T cells either induced by mRNA vaccination (still including the ancestral spike sequence) or by natural infection (occurring in the pre-omicron period) cross-recognize omicron due to conservation of the targeted epitopes. This experimental approach is different but clearly complementary to the approach comparing overall T cell responses reactive against omicron or ancestral virus as suggested by reviewer #3. In the revised version of our manuscript, we have better clarified our experimental approach

e.g. on page 2, lines 45-46 (abstract), page 4, lines 116-117, and page 5, lines 165-166. Hence, we believe that our main conclusion “*Broad targeting of omicron by SARS-CoV-2- specific T cells after infection and vaccination*” is supported by our experimental approach and

relevantly complements the important findings gained by testing the overall cross-reactive T cell response towards SARS-CoV-2 omicron. An additional advantage of our approach is that our results can be easily transferred to newly emerging variants of concern (VOCs). Indeed, since the omicron subtype BA.2 already dominates the earlier omicron “prototype” BA.1 subtype in many countries, we have now extended our analysis to this subtype (see especially novel Extended Data Fig. 1B, Extended Data Fig. 4B, and Supplementary Table 3).

In addition, according to this reviewer’s comment, we and others showed that a broad T cell response targeting epitopes in different viral proteins is associated with a mild course of COVID-19. As depicted in Figure 1A, our data confirm the broad targeting of multiple SARS-CoV-2 proteins in convalescents. We have more strongly emphasized this point in the revised version of the manuscript (page 2/3, lines 67-69). In this study, we analyzed the spike-specific T cell epitope repertoire after mRNA vaccination compared to natural infection in more detail and our conclusions concerning the comprehensively analyzed repertoire of the T cell response are indeed limited to the spike protein. We have clarified this point throughout the revised manuscript.

As suggested by this reviewer, we have added the comparison of the intensity of T cell responses (indicated as %IFN-gamma producing T cells) in the two groups as Extended data Fig. 1A to the revised manuscript. These data again highlight the important role of non-spike epitopes in resolved SARS-CoV-2 infection.

Please find below the detailed comments on the manuscript and figures:

- Of the 19 convalescent donors three were also vaccinated. Did the authors include samples of those donors before vaccination? Otherwise, these three should not be included in the cohort of convalescents and only used for the longitudinal analysis.

In the analysis of the cohort of 19 convalescent donors (e.g., Fig. 1 and associated Extended Data Fig. 1), we included pre-vaccination samples of these three donors. Samples after vaccination of these three patients, however, were included in Fig. 2B and Extended Data Fig. 3B. These figure panels also contain a timeline of sampling in these three patients. Following the next comment by the reviewer, we have now also included the exact sampling time points in Supplementary Table 1. This will definitively help to clarify the important issue of timing of sampling, and we apologize for any ambiguity in the previous version of our manuscript.

- At which time point after vaccination or after infection were the blood samples collected? That should clearly be stated e.g. in Sup Table 1.

The time points were added accordingly in Supplementary Table 1.

- The authors should provide a table with the 43 tested peptide sequences. In the method section, 60 peptides were described. Where does the difference between 43 and 60 come from?

We now provide a Table with the peptide sequences as requested (Supplementary Table 2). The document can also be found on the public repository “Open Science Framework” via <https://osf.io/zbk6q/>. Of note, we tested a total of 60 peptides, but included only those 43 peptides in the analysis (e.g., Fig. 1A) that were tested in a minimum of 3 patients with the corresponding HLA type for statistical reasons. We apologize for this ambiguity in the previous version of our manuscript and now explain this selection strategy in the methods section (page 7, lines 207-210 of the revised manuscript).

- Regarding the overlapping peptides: line 105: 180 peptides, method section line 29: 182 peptides, method section line 38: 181 peptides. What is the correct number?

We apologize for this inconsistency in the previous version of the manuscript. We designed the set of overlapping peptides to include 180 peptides with the ancestral virus sequences plus 2 peptides containing the 2 amino acid substitutions present in the “stabilized” mRNA vaccine sequence. We now explain the design and number of overlapping peptides on page 7, lines 204-207 of the revised manuscript and use the number of 182 overlapping peptides throughout the manuscript.

- Regarding the one spike-derived peptide affected by omicron (red in Figure 1A), a supplementary Figure or Table showing the sequence of the peptide and the sequence of the omicron variant would be helpful. Is it only affected by one amino acid? In which position? Anker position? The same could be analyzed for the affected peptides in Figure 1B and D.

This important information is now displayed in Supplementary Table 3, taking into account the omicron subtypes BA.1 and BA.2.

- Why did the authors not test the convalescent and the vaccinated donors with the omicron peptides? Only by analyzing the T cell response against ancestral peptides AND omicron peptides can truly demonstrate if the T cells are able to recognize omicron sequences or if the T cell response in total is not reduced or affected by omicron.

As mentioned above, we completely agree with this reviewer about the relevance to test the omicron-reactive T cell response to address whether the T cell response in total is affected by omicron. Our approach is complementary, focusing on the identification of T cell responses elicited by ancestral spike sequences in mRNA vaccination compared to natural infection in the pre-omicron times down to the epitope level. Hence, we do not measure cross-reactive but rather cross-recognizing T cell responses towards omicron. We have thoroughly used the term “cross-recognizing” in the revised manuscript.

- Figure 1: All the Figures should be labeled with “convalescents (n=19)” and “vaccinees (n = 16)”. Do the authors show only vaccinees after the second dose? This should be clearly labeled within the figure and in the legend. Furthermore, the figure legend is not sufficient enough to understand the figure e.g. explanation of wt vs mut in B and D or abbreviations such as NTD, RBD, FCS are missing.

The Figures are now labeled accordingly to clarify which cohort is depicted. In Fig. 1, only vaccinees after the second dose were analyzed, and this is now clearly stated in the legend. The abbreviations NTD, RBD, and FCS as well as the exact terminology of wt versus mut are now explained in the legend to Fig. 1.

- Figure 1A: Are the frequency of donors who recognize a peptide normalized to the HLA allotypes? If not, why did 100% of convalescents recognize A*01-restricted peptides where only 6/19 convalescents are A*01 positive and why did 100% of the vaccinees recognize the A*03-restricted peptide where only 6/16 vaccinees are A*03 positive. How do the authors explain this?

The frequency of donors who recognize a peptide is indeed normalized to the HLA allotypes. This is now explained in the legend to Fig. 1A.

- Figure 1B/D: Are the n numbers behind the HLA allotypes the number of donors with this allotype? One summary line per plot summarizing all donors independent of the allotype would be helpful.

Indeed, the numbers indicate the number of individuals with the respective HLA allotype. This is now stated in the legend to Fig. 1B. We have added a summary line per plot summarizing all donors as suggested.

- Figure 1F: For the convalescents CD4+ only 18 points are visible. That should be 19, shouldn't it? Regarding the statistics: A paired t test is not correct here since the donors of the different cohorts are not paired. I would suggest a nonparametric test such as the Mann-Whitney-Test.

We apologize for the missing data point in previous Fig. 1F, now Extended Data Fig. 1E. We have now included the missing data. In addition, we have used the nonparametric Mann-Whitney-Test as suggested in the revised version (Extended Data Fig. 1E).

- Did the authors test the donors of the vaccinee cohort for anti-nucleocapsid antibodies to be sure they did not have a previously unknown SARS-CoV-2 infection?

The vaccinees were now tested for anti-nucleocapsid antibodies to exclude a previously unknown SARS-CoV-2 infection. This important information is now stated on page 6, lines 184-185 of the methods section.

- The data in the extended data Figure 1 does not match the statement in the text (line 112/113). The authors discuss the differences between convalescent and vaccinees and not the longitudinal analysis within the different cohorts, which is shown in the figure.

We agree with the reviewer that the longitudinal analysis was not mentioned at the best suited text paragraph. We now mention the stable T cell epitope repertoire observed in the longitudinal analysis (now Extended Data Fig. 2) on page 4, lines 106-108.

- Extended Data 1B: The different donors in these plots should be clearly labeled. The number of T cell responses depends on the HLA type of the donors hampering a comparison over all donors in the same plot.

In previous Extended Data Fig. 1B (now Extended Data Fig. 2B), the number of targeted overlapping peptides is indicated, irrespective of HLA type of the individual. All 19 convalescents are included in this analysis, and this is now clearly stated in the figure legend.

- Using the terms cross-reactive and spike-specific together is very confusing and mutually exclusive (line 132/140). Cross-reactive peptides can be recognized by non-SARS-CoV-2-specific common cold coronavirus-primed T cells. The authors did not know if the vaccinees have cross-reactive T cells targeting the spike peptides before the vaccination.

We have removed the ambiguous term cross-reactive throughout the manuscript.

- Figure 2: Did the authors test the data for normality before using a t test? Maybe a nonparametric test such as Wilcoxon matched-pairs signed rank test would be better.

We have now used the nonparametric Wilcoxon matched-pairs signed rank test in the revised version of Fig. 2.

- The data about the sweep regions are not convincing since omicron showed 7 mutations in these regions.

As also pointed out in the reply to reviewer #1, selective sweep regions were defined according to Kang *et al.* (Kang L, He G, Sharp AK, et al. A selective sweep in the Spike gene has driven SARS-CoV-2 human adaptation. *Cell* 2021;184:4392-400). In brief, Kang et al. analyzed total of 136,114 complete SARS-CoV-2 genomes from the human host. Subsequently to alignment with MAFFT, sweep regions were detected by using OmegaPlus and RAiSD, with the original SARS-CoV-2 isolate Wuhan-Hu-1 genome (NC_045512.2) as an outgroup. The common

outliers were manually grouped into eight regions of at least 50 bp, four of these were located in ORF1ab and four in the S region. Selective sweep 1 S₃₂₃₋₄₃₄, selective sweep 2 S₅₂₄₋₅₄₅, selective sweep 3 S₈₈₈₋₉₁₉, selective sweep 4 S_{965-1,050} were identified by this method. VOC were aligned to these selective sweep regions and peptides were mapped to the spike protein, to identify peptides that localize to the selective sweep regions identified by Kang *et al.*. Hence, selective sweep regions were not only defined based on the currently circulating VOC but more broadly cover SARS-CoV-2 genomes. We have extended the information about the definition of the selective sweep regions in the Methods section (page 8, lines 253-263 of the revised manuscript) and now more clearly state that the selective sweep regions are affected by point mutations in omicron (page 5, lines 145-147 of the revised manuscript).

- The extended data Figure 3 is not referred to in the text. For Pacific Blue the stained marker CD4 is not labeled on the axis.

We have now referred to the Extended Data (now Extended Data Fig. 5) in the Methods section (page 8, line 240) and included the missing label for CD4 in the revised Extended data Fig. 5.

- Some typos: line 1: SARS-CoV-2-specific T cells, line 71: breakthrough, Supplementary Table 1: A*30:04 for donor 35

We have corrected the typos in the revised version of the manuscript.

- Supplementary Table 1: List of abbreviations is missing. Mild defines the course of COVID-19 not the natural infection.

We have added a list of abbreviations and have changed the title of the respective column in Supplementary Table 1 from "Natural infection" to "Severity of COVID-19"

Decision Letter, first revision:

Dear Maike,

Thank you for submitting your revised manuscript "Broad targeting of omicron by SARS-CoV-2-specific T cells after infection and vaccination" (NMICROBIOL-22010115A). It has now been seen by the original referees and their comments are below. The reviewers find that the paper has improved in revision, and therefore we'll be happy in principle to publish it in Nature Microbiology, pending minor revisions to satisfy the referees' final requests and to comply with our editorial and formatting guidelines. In addition, just to let you know, we have changed the article type for the submission to Article as this will give you more space to include all required data, information and references as compared to a Brief Communication.

Thank you again for your interest in Nature Microbiology Please do not hesitate to contact me if you have any questions.

Sincerely,

{redacted}

Reviewer #1 (Remarks to the Author):

The authors comments to the reviewers concerns have been sufficiently addressed.

Reviewer #2 (Remarks to the Author):

All my concerns have been addressed

Reviewer #3 (Remarks to the Author):

The authors have implemented the suggestions well and significantly improved the quality of the manuscript. However, the title of the manuscript is still not reflecting the performed experiments and received results! The title should be changed and not include omicron!

The following small errors have been noticed:

- Supplementary Table 3: "Small IC50/low rank": The authors should better explain what the cutoff values are for strong, weak and no binders.
- Figure 1B: the n numbers of convalescents and vaccinees are interchanged in the CD8 plots. In the Figure legend the authors should add "to overlapping peptides of the spike protein"
- Figure 2A and B: The wt legend is not clearly labeled.

Decision Letter, final checks:

Dear Maïke,

Thank you for your patience as we've prepared the guidelines for final submission of your Nature Microbiology manuscript, "Broad targeting of omicron by SARS-CoV-2-specific T cells after infection and vaccination" (NMICROBIOL-22010115A). Please carefully follow the step-by-step instructions provided in the attached file, and add a response in each row of the table to indicate the changes that you have made. Please also check and comment on any additional marked-up edits we have proposed within the text. Ensuring that each point is addressed will help to ensure that your revised manuscript can be swiftly handed over to our production team.

In recognition of the time and expertise our reviewers provide to Nature Microbiology's editorial process, we would like to formally acknowledge their contribution to the external peer review of your manuscript entitled "Broad targeting of omicron by SARS-CoV-2-specific T cells after infection and vaccination". For those reviewers who give their assent, we will be publishing their names alongside the published article.

Nature Microbiology offers a Transparent Peer Review option for new original research manuscripts submitted after December 1st, 2019. As part of this initiative, we encourage our authors to support increased transparency into the peer review process by agreeing to have the reviewer comments, author rebuttal letters, and editorial decision letters published as a Supplementary item. When you submit your final files please clearly state in your cover letter whether or not you would like to participate in this initiative. Please note that failure to state your preference will result in delays in accepting your manuscript for publication.

Cover suggestions

As you prepare your final files we encourage you to consider whether you have any images or illustrations that may be appropriate for use on the cover of Nature Microbiology.

Nature Microbiology has now transitioned to a unified Rights Collection system which will allow our Author Services team to quickly and easily collect the rights and permissions required to publish your work. Approximately 10 days after your paper is formally accepted, you will receive an email in providing you with a link to complete the grant of rights. If your paper is eligible for Open Access, our Author Services team will also be in touch regarding any additional information that may be required to arrange payment for your article.

Please note that *Nature Microbiology* is a Transformative Journal (TJ). Authors may publish their research with us through the traditional subscription access route or make their paper immediately open access through payment of an article-processing charge (APC). Authors will not be required to make a final decision about access to their article until it has been accepted. [Find out more](https://www.springernature.com/gp/open-research/transformative-journals)

about Transformative Journals

Please use the following link for uploading these materials:
{redacted}

Best regards,

{redacted}

Reviewer #1:

Remarks to the Author:

The authors comments to the reviewers concerns have been sufficiently addressed.

Reviewer #2:

Remarks to the Author:

All my concerns have been addressed

Reviewer #3:

Remarks to the Author:

The authors have implemented the suggestions well and significantly improved the quality of the manuscript. However, the title of the manuscript is still not reflecting the performed experiments and received results! The title should be changed and not include omicron!

The following small errors have been noticed:

- Supplementary Table 3: "Small IC50/low rank": The authors should better explain what the cutoff values are for strong, weak and no binders.
- Figure 1B: the n numbers of convalescents and vaccinees are interchanged in the CD8 plots. In the Figure legend the authors should add "to overlapping peptides of the spike protein"
- Figure 2A and B: The wt legend is not clearly labeled.

Final Decision Letter:

Dear Maïke,

I am pleased to accept your Article "SARS-CoV-2-specific T cell epitope repertoire in convalescent and mRNA-vaccinated individuals" for publication in Nature Microbiology. Thank you for having chosen to submit your work to us and many congratulations.

Due to the importance of these deadlines, we ask you to please let us know now whether you will be difficult to contact over the next month. If this is the case, we ask you to provide us with the contact information (email, phone and fax) of someone who will be able to check the proofs on your behalf, and who will be available to address any last-minute problems.

Acceptance of your manuscript is conditional on all authors' agreement with our publication policies (see <https://www.nature.com/nmicrobiol/editorial-policies>). In particular your manuscript must not be published elsewhere and there must be no announcement of the work to any media outlet until the publication date (the day on which it is uploaded onto our website).

Please note that *Nature Microbiology* is a Transformative Journal (TJ). Authors may publish their research with us through the traditional subscription access route or make their paper immediately open access through payment of an article-processing charge (APC). Authors will not be required to make a final decision about access to their article until it has been accepted. [Find out more about Transformative Journals](https://www.springernature.com/gp/open-research/transformative-journals)

Authors may need to take specific actions to achieve [compliance with funder and institutional open access mandates](https://www.springernature.com/gp/open-research/funding/policy-compliance-faqs). If your research is supported by a funder that requires immediate open access (e.g. according to [Plan S principles](https://www.springernature.com/gp/open-research/plan-s-compliance)) then you should select the gold OA route, and we will direct you to the compliant route where possible. For authors selecting the subscription publication route, the journal's standard licensing terms will need to be accepted, including [self-archiving policies](https://www.springernature.com/gp/open-research/policies/journal-policies). Those licensing terms will supersede any other terms that the author or any third party may assert apply to any version of the manuscript.
